# Self-Rule to Adapt: Learning Generalized Features from Sparsely-Labeled Data Using Unsupervised Domain Adaptation for Colorectal Cancer Tissue Phenotyping

**Christian Abbet**[1,2]                                    CHRISTIAN.ABBET@EPFL.CH
[1] *Signal Processing Laboratory 5 (LTS5), EPFL, Lausanne, Switzerland*
[2] *Institute of Pathology, University of Bern, Switzerland*

**Linda Studer**[2,3,4]                                    LINDA.STUDER@UNIFR.CH
[3] *Document, Image and Video Analysis (DIVA) Research Group, University of Fribourg, Switzerland*
[4] *iCoSyS, University of Applied Sciences and Arts Western Switzerland, Switzerland*

**Andreas Fischer**[3,4]                                    ANDREAS.FISCHER@HEFR.CH
**Heather Dawson**[2]                              HEATHER.DAWSON@PATHOLOGY.UNIBE.CH
**Inti Zlobec**[2]                                      INTI.ZLOBEC@PATHOLOGY.UNIBE.CH
**Behzad Bozorgtabar**[1,5]                             BEHZAD.BOZORGTABAR@EPFL.CH
[5] *Center for Biomedical Imaging (CIBM), Switzerland*

**Jean-Philippe Thiran**[1,5,6,7]                       JEAN-PHILIPPE.THIRAN@EPFL.CH
[6] *University of Lausanne (UNIL), Switzerland*
[7] *Radiology Department, Centre Hospitalier Universitaire Vaudois (CHUV), Switzerland*

## Abstract

Supervised learning is constrained by the availability of labeled data, which are especially expensive to acquire in the field of digital pathology. Making use of open-source data for pre-training or using domain adaptation can be a way to overcome this issue. However, pre-trained networks often fail to generalize to new test domains that are not distributed identically due to variations in tissue stainings, types, and textures. Additionally, current domain adaptation methods mainly rely on fully-labeled source datasets. In this work, we propose Self-Rule to Adapt (SRA) which takes advantage of self-supervised learning to perform domain adaptation and removes the necessity of a fully-labeled source dataset. SRA can effectively transfer the discriminative knowledge obtained from a few labeled source domain to a new target domain without requiring additional tissue annotations. Our method harnesses both domains' structures by capturing visual similarity with intra-domain and cross-domain self-supervision. We show that our proposed method outperforms baselines for domain adaptation of colorectal tissue types and further validate our approach on our in-house clinical cohort.

**Keywords:** Computational pathology, self-supervised learning, few labeled data, unsupervised domain adaptation, colorectal cancer

## 1. Introduction

Colorectal cancer (CRC) is one of the most common cancers worldwide and its understanding through computational pathology techniques can significantly improve the chances of effective treatment (Smit and Mesker, 2020) by refining disease prognosis and assisting

pathologists in their daily routine. The data used in computational pathology most often consists of Hematoxylin and Eosin (H&E) stained whole slide images (WSIs) (Hegde et al., 2019; Lu et al., 2020) and tissue microarrays (TMAs) (Nguyen et al., 2021)

Although fully supervised deep learning models have been widely used for a variety of tasks, including tissue classification (Kather et al., 2019) and semantic segmentation (Qaiser et al., 2019), in practice it is time-consuming and expensive to obtain fully-labeled data as it involves expert pathologists. This hinders the applicability of supervised machine learning models to real-world scenarios. Self-supervised learning was proposed to address these limitations. It involves a two-step training scheme, where "*data creates its own supervision*"(Pieter et al., 2020) to learn rich features from structured unlabeled data and to create supervision from itself. Applications of this approach in computational pathology include survival analysis (Abbet et al., 2020) and WSIs classification (Li et al., 2020).

In addition, different techniques such as stain normalization (Macenko et al., 2009) algorithms and unsupervised domain adaptation (UDA) methods have been developed with the aim of improving the classification of heterogeneous WSIs. UDA methods address this issue by learning from a rich source domain together with the label-free target domain to have a well-performing model on the target domain at inference time. DANN (Ganin and Lempitsky, 2015) for example uses gradient reversal layers, to learn domain-invariant features. Self-Path (Koohbanani et al., 2020) combines the DANN approach and self-supervised auxiliary tasks such as the hematoxylin prediction to improve stability.

Another example is CycleGAN (Zhu et al., 2017), which takes advantage of adversarial learning to cyclically map images between the source and target domain. However, adversarial approaches can fall short, because they do not consider task-specific decision boundaries, and only try to distinguish the features as either coming from the source or target domain (Saito et al., 2018a).

A further issue is that most methods treat the domain adaptation as a closed-set scenario, which assumes that all target samples belong to a class present in the source domain, even though this is often not the case. To overcome this OSDA (Saito et al., 2018b) proposes an adversarial open-set domain adaptation approach, where the feature generator has the option to reject mistrusted target samples as an additional class. Another recent work SSDA (Xu et al., 2019) uses self-supervised domain adaptation methods that combine auxiliary tasks such as image rotation or jigsaw puzzle-solving, adversarial loss, and batch normalization calibration across source and target domains.

In this work, we propose a label-efficient framework called Self-Rule to Adapt (SRA) for tissue type recognition in histological images and attempt to overcome the above-mentioned issues by combining self-supervised learning approaches with UDA. We present an entropy-based approach that progressively learns domain invariant features thus making our model more robust to class definition inconsistencies as well as the presence of unseen tissue classes when performing domain adaptation. SRA is able to accurately classify and segment tissue types in H&E stained images, which is an important step for many downstream tasks. Our proposed method achieves this by making use of few labeled open-source datasets as well as unlabeled data, that are abundant in digital pathology, reducing the annotation workload for pathologists. We show that our method outperforms previous domain adaptations approaches in a few-label setting and its potential for clinical application in the diagnostics of CRC.

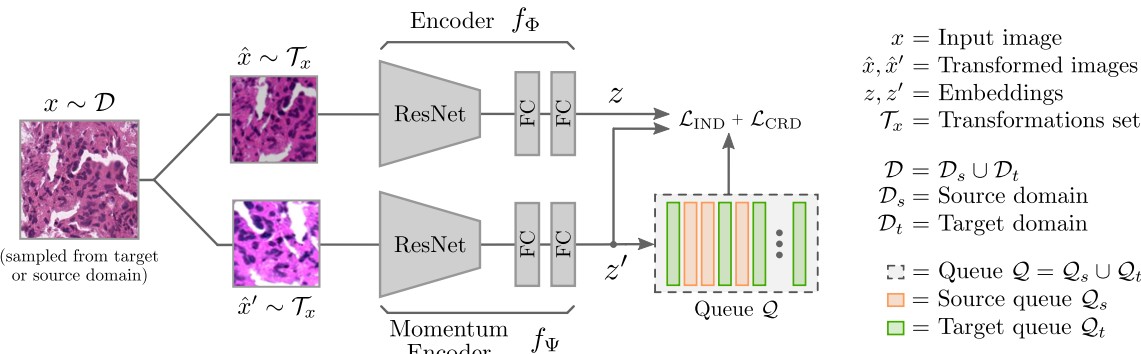

Figure 1: The proposed Self-Rule to Adapt (SRA) architecture for a given input image $x$. The $\mathcal{L}_{\text{IND}}$ and $\mathcal{L}_{\text{CRD}}$ represent the in-domain and cross-domain loss respectively.

## 2. Methods

In our unsupervised domain adaptation setting, we have access to a small set of labeled source data, sampled from a source domain distribution and a set of unlabeled target data from a target distribution. The goal is to learn a hypothesis function (e.g., classifier here) on the source domain that provides a good generalization in the target domain. To this end, we propose a novel self-supervised cross-domain adaptation setting, which is described in more detail below. Figure 1 gives an overview of the proposed network architecture.

Our model builds upon two networks $f_\Phi$, $f_\Psi$ that compute the query $z$ and key $z'$ embedding from the input representations $\hat{x}$, $\hat{x}'$, respectively. Each branch consists of a residual encoder and two fully connected layers based on the state-of-the-art (SOTA) architecture proposed in Chen et al. (2020b). To generate $\hat{x}$, $\hat{x}'$, a random image $x$ is drawn from either the source $\mathcal{D}_s$ or the target $\mathcal{D}_t$ domain and is then transformed with two random data augmentations selected from $\mathcal{T}_x$ to create a matching pair. The key embeddings $z'$ are used to maintain a queue $\mathcal{Q}$ of negative samples $\{q_i\}_{i=1}^{|\mathcal{Q}|} \in \mathcal{Q}$ in a first-in, first-out fashion. The queue provides a large number of examples which alleviates the need for a large batch (Chen et al., 2020a) or the use of memory banks (Kim et al., 2020). Moreover, $f_\Psi$ is updated using a momentum approach, combining its weights to those of $f_\Phi$. This approach ensures that $f_\Psi$ generates a slowly-shifting embedding. Motivated by Ge et al. (2020); Kim et al. (2020), we extend the domain adaptation learning procedure to our model definition and task. Hence, we split the loss terms into two distinct tasks, namely in-domain $\mathcal{L}_{\text{IND}}$ and cross-domain $\mathcal{L}_{\text{CRD}}$ representation learning. The objective loss $\mathcal{L}_{\text{SRA}} = \mathcal{L}_{\text{IND}} + \mathcal{L}_{\text{CRD}}$ is the summation of both terms, and are described in more detail below.

### 2.1. In-domain Loss

The first objective $\mathcal{L}_{\text{IND}}$ aims at learning the individual distribution of each the source and the target domain features. We want to keep the two domains independent as we will optimize their alignment later. For each vector $z$, there is a paired embedding $z'$ that is generated from the same tissue image and therefore is, by definition, similar. The contrastive

loss, as expressed in Equations (1) and (2), is therefore used to constrain the representation of the embedding space for each domain separately.

$$p_i^{\text{IND}}(\mathcal{Q}) = \frac{\exp(z_i^\top z_i'/\tau)}{\exp(z_i^\top z_i'/\tau) + \sum_{l \in \mathcal{Q}} \exp(z_i^\top q_l/\tau)}. \tag{1}$$

$$\mathcal{L}_{\text{IND}} = \frac{-1}{|D_s| + |D_t|} \left( \sum_{i \in D_s} \log \left[ p_i^{\text{IND}}(\mathcal{Q}_s) \right] + \sum_{i \in D_t} \log \left[ p_i^{\text{IND}}(\mathcal{Q}_t) \right] \right). \tag{2}$$

We denote $\mathcal{Q}_s, \mathcal{Q}_t \subset \mathcal{Q}$ as the sets of indexed samples of the queue that are drawn from the corresponding domain $\mathcal{D}_s, \mathcal{D}_t$, and $\tau \in \mathbb{R}$ as the temperature. The temperature is typically small ($< 1$) to help the model in making confident predictions. For all images of each dataset $\mathcal{D}_s, \mathcal{D}_t$, we want to minimize the distance between $z$ and $z'$ while maximizing the distance to the previously generated negative samples from the corresponding set $\mathcal{Q}_s, \mathcal{Q}_t$. The queue samples are considered reliable negative candidates as they are generated by $f_\Psi$ whose weights slowly varies due to its momentum update procedure.

## 2.2. Cross-domain Loss

We can see the cross-domain matching task as the generation of features that are discriminative for both sets. In other words, if we embed a random sample drawn from $\mathcal{D}_s$ we expect to be able to find a limited number of candidates in $\mathcal{D}_t$ whose representations contain similar information as our initial query. Based on this logic, we compute the similarities and entropy of a query sample $z_i$ drawn from one set (for example $\mathcal{D}_s$) and the stored queue samples from the other set (for example $\mathcal{Q}_t$):

$$H_i^{\text{CRD}}(\mathcal{Q}) = -\sum_{j \in \mathcal{Q}} p_{i,j}^{\text{CRD}}(\mathcal{Q}) \log \left[ p_{i,j}^{\text{CRD}}(\mathcal{Q}) \right] \quad \text{and} \quad p_{i,j}^{\text{CRD}}(\mathcal{Q}) = \frac{\exp(z_i^\top q_j/\tau)}{\sum_{l \in \mathcal{Q}} \exp(z_i^\top q_l/\tau)}. \tag{3}$$

Low entropy means that the selected query from one domain matches with a limited number of keys from another domain. The loss, therefore, aims to minimize the average entropy of the similarity distributions, assisting the model in making confident predictions:

$$\mathcal{L}_{\text{CRD}} = \frac{1}{|D_s| + |D_t|} \left[ \sum_{i \in D_s} H_i^{\text{CRD}}(\mathcal{Q}_t) + \sum_{i \in D_t} H_i^{\text{CRD}}(\mathcal{Q}_s) \right]. \tag{4}$$

## 2.3. Easy-to-hard (E2H) Learning

At the start of the learning process, the correlation between samples and their entropy is unclear as the model weights are initialized randomly, which does not guarantee proper feature descriptors. Additionally, being able to find matching samples for all input queries across datasets is a strong assumption. In clinical applications, we often rely on open-source datasets with a limited number of classes to annotate complex tissue databases. For example, tissues coming from specific cancer subtypes, such as mucinous CRC, might not be present in a public dataset while being potentially frequent in daily diagnostics. In other words, optimizing Equation (3) will result in a performance drop as the loss will try to find cross-domain candidates even if there are none to be found.

To tackle this issue, we introduce an easy-to-hard learning scheme. We start with easy (low entropy) samples and progressively include harder (high entropy) samples as the training progresses. We substitute the summation over $\mathcal{D}_s, \mathcal{D}_t$ in Equation (4) with the corresponding set of candidates $\mathcal{R}_s, \mathcal{R}_t$ defined in Equation (5) where the ratio $0 \leq r \leq 1$ is gradually updated during training using a step function. We denote $s_w, s_h$ as the width and height of the step respectively.

$$\mathcal{R}_{s/t} = \{i \in \mathcal{D}_{s/t} \mid H_i^{\mathrm{CRD}}(\mathcal{Q}_{t/s}) \text{ is reverse top-}r\} \quad \text{and} \quad r = \left\lfloor \frac{\text{epoch}}{N_{\text{epochs}} \cdot s_w} \right\rfloor \cdot s_h, \quad (5)$$

## 3. Results and Discussion

In this section, we present the results of the experiments. We validate our proposed self-supervised domain adaptation approach on two publicly available datasets (Section 3.1) and compare it to current SOTA methods for UDA in Section 3.2. To assess the performance of our approach in a clinically-relevant use case, we further validate it on WSIs crops from our in-house cohort in Section 3.3. To help future research, the implementation is open source[1]. Further details on the experimental setup can be found in Appendix A.

### 3.1. Public Datasets

In this study, we use two publicly available datasets, Kather-19 (K19) (Kather et al., 2019) and Kather-16 (K16) (Kather et al., 2016). The former is composed of $100,000$ image patches sampled from 9 different CRC tissue types (tumor, stroma, muscle, lymphocytes, debris, mucus, normal mucosa, adipose, and background) while the latter includes $5,000$ crops distributed over 8 tissue types (tumor, stroma, complex stroma, lymphocytes, debris, normal mucosa, adipose, background). Following a discussion with expert pathologists, we group stroma/muscle and debris/mucus as stroma and debris respectively to create a corresponding adaptation between K19 and K16. Complex stroma who is only present in K16 is kept for training but excluded from the evaluation process. With this problem definition, we fall into an open set scenario where the class distribution of the two domains does not rigorously match, as opposed to a closed set adaptation scheme. For more information about the datasets and their discrepancies please refer to Appendix B.

### 3.2. Cross-Domain Patch Classification

In this task, we use the larger set K19 with 1% of the source labels available and adapt it to K16 in order to simulate the clinical application where we usually rely on a large quantity of unlabeled data and only have access to few labeled samples. The results of our proposed SRA method are presented in Table 1, in comparison with the SOTA algorithms for domain adaption. We first train our model in an unsupervised fashion ($\mathcal{L}_{\mathrm{SRA}}$) and then fit a linear classifier with few source labels on top of the frozen model weights. As the lower bound, we consider direct transfer learning, where the model is trained in a supervised fashion on the source data only. We use the same logic for the upper bound by training on the

---

1. Code available on https://github.com/christianabbet/SRA.

Table 1: Results of the domain adaptation from Kather-19 (source) to Kather-16 (target). 1% of the source domain labels are known and the target domain labels are unknown. The top results for the domain adaptation methods are highlighted in bold. We report the F1 score for each class, as well as the overall F1 score.

| Methods | TUM | STR | LYM | DEB | NORM | ADI | BACK | ALL |
|---|---|---|---|---|---|---|---|---|
| Source only[‡] | 74.0[**] | 77.4[**] | 75.3[**] | 50.5[**] | 66.9[**] | 87.0[**] | 93.1[**] | 75.1[**] |
| Stain norm. (Macenko et al., 2009) | 77.8[**] | 75.9[**] | 68.2[**] | 42.1[**] | 75.1[**] | 77.4[**] | 87.6[**] | 72.2[**] |
| CylceGAN (Zhu et al., 2017) | 70.7[**] | 71.6[**] | 62.3[**] | 47.6[**] | 75.5[**] | 89.0[**] | 88.2[**] | 72.4[**] |
| DANN (Ganin and Lempitsky, 2015) | 65.8[**] | 60.8[**] | 42.3[**] | 47.8[**] | 61.9[**] | 64.1[**] | 62.3[**] | 57.8[**] |
| SelfPath (Koohbanani et al., 2020) | 71.5[**] | 68.8[**] | 68.1[**] | 57.6[**] | 77.6[**] | 82.3[**] | 85.5[**] | 73.1[**] |
| OSDA (Saito et al., 2018b) | 82.0[**] | 78.2[*] | **83.6** | 63.8[**] | 80.3[**] | 90.8[**] | 93.2[*] | 81.7[**] |
| SSDA - Jigsaw (Xu et al., 2019) | 90.0[**] | **81.2** | 79.5[**] | 64.4[**] | 88.3[**] | 94.2[**] | 95.7[*] | 84.9[**] |
| SRA (ours) | **93.4** | 72.9[**] | **82.7**[+] | **67.9** | **96.5** | **97.0** | **97.2** | **86.9** |
| Target only[†] | 94.6[+] | 83.6[**] | 92.6[**] | 88.7[**] | 95.4[+] | 97.8[*] | 98.5[*] | 93.0[**] |

[‡] Direct transfer learning: trained on the source domain only, no adaptation (lower bound).
[†] Fully supervised: trained knowing all labels of the target domain (upper bound).
[+] $p \geq 0.05$; [*] $p < 0.05$; [**] $p < 0.001$; unpaired t-test with respect to the top result.

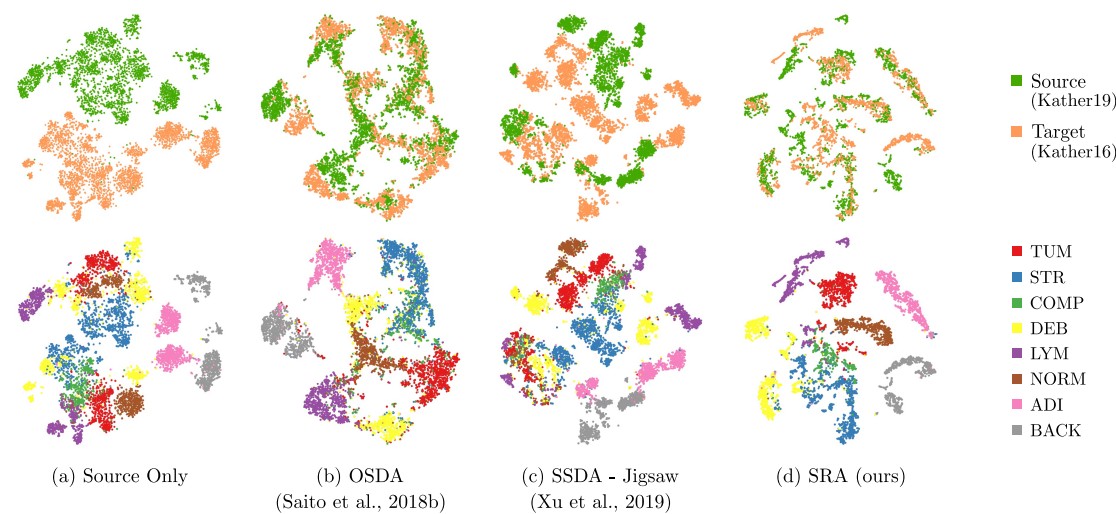

(a) Source Only    (b) OSDA (Saito et al., 2018b)    (c) SSDA - Jigsaw (Xu et al., 2019)    (d) SRA (ours)

Figure 2: The t-SNE projection of the source (Kather-19) and target (Kather-16) domain embeddings. The top row shows the alignment between the source and target domain, while the bottom row highlights the representations of the different classes. We compare our approach (d) to other UDA methods (a-c).

target domain data (fully supervised approach). Figure 2 shows the t-SNE projection and alignment of the domain adaptation for the source only, top-performing baselines (OSDA, SSDA with jigsaw solving), and our method (SRA). Appendix C-E provides the results of the ablation study as well as additional results.

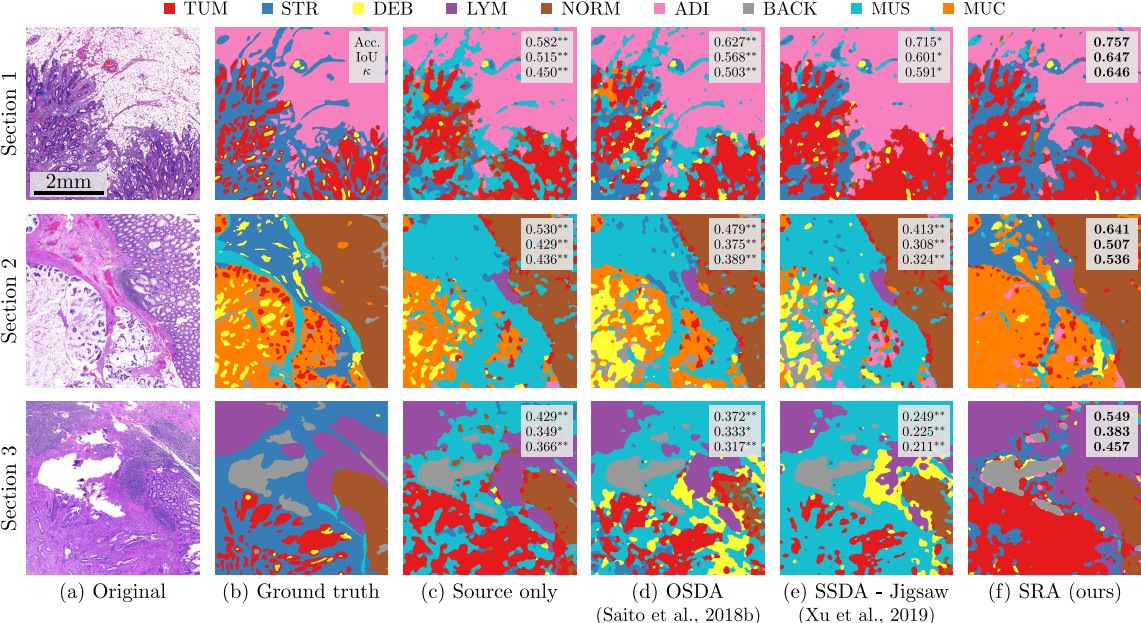

+ $p \geq 0.05$;  $^*p < 0.05$;  $^{**}p < 0.001$; unpaired t-test with respect to the top result.

Figure 3: Examples of domain adaptations from K19 to our in-house dataset. (a-b) show the original sections from the WSIs and their ground truth, respectively. We compare the performance of our Self-Rule to Adapt (SRA) algorithm (f) to the lower bound and the previous top methods (c-e). We report the pixel-wise accuracy, the weighted intersection over union, and the pixel-wise Cohen's kappa ($\kappa$) score.

Stain normalization slightly decreases the performance as it introduces color artifacts that trick the network classifier. Our proposed SRA method shows an excellent alignment between the same class clusters of the source and target distributions and outperforms other approaches in terms of weighted F1 score. Notably, our approach is even able to match the upper bound model for normal and tumor tissue identification. The embedding of complex stroma, which only exists in the target domain, is represented as a single cluster with no matching candidates, which shows that the model was not forced to find suitable matches. Furthermore, the cluster representation is more compact compared to SSDA, where for example normal mucosa tends to be aligned with complex stroma and tumor. SSDA and OSDA misclassify debris as lymphocytes due to their similar texture and structure. Self-Path suffers from DANN who's loss is unstable leading to large performances gaps when training. Heavier data augmentations partially solved this issue. Our approach suffers a drop in performance for stroma detection, which can be explained by the presence of lymphocytes in numerous stroma tissue examples, causing a higher rate of misclassification.

### 3.3. Use Case: Cross-Domain Segmentation of WSIs

To further validate our approach in a real case scenario, we perform domain adaptation using our proposed model from K19 to our in-house slides and validate it on WSIs sections.

We randomly extract patches from over 300 WSIs to train our model and then use a sliding window approach to predict the class of each patch in the selected regions. The final prediction map is smoothed using conditional random fields as in Chan et al. (2019). The results are presented in Figure 3, alongside the original H&E crop, their corresponding segmentation annotated by an expert pathologist according to the definitions used in the K19 dataset, as well as comparative results of the introduced approaches. The three sections were selected such that, overall, they represent all tissue types equally.

Our approach outperforms the domain adaptation methods in terms of pixel-wise accuracy, weighed intersection over union (IoU) and pixel-wise Cohen's kappa score. For all models, stroma and muscle are poorly differentiated as both have similar visual features. This phenomenon is even more apparent in the source only setting where muscle tissue is almost systematically interpreted as stroma. SSDA tends to predict lymphocyte aggregates as debris, which can be explained by its sensitivity to staining variations. OSDA on the other side fails to adapt and generalize to new debris examples while trying to reject mistrusted samples. Regions with mixtures of tissue types (e.g., lymphocytes + stroma or stroma + isolated tumor cells) are challenging cases because the samples available in online cohorts mainly contain ideal examples with homogeneous tissue textures for each patch, and no mixed class examples. Subsequently, domain adaptations naturally struggle to align features resulting in a biased classification. We also observe that thinner or torn stroma regions, where the background behind is well visible, are often misclassified as adipose tissue by SRA, which is most likely due to their similar appearance. However, our SRA model is able to correctly distinguish between normal mucosa and tumor, which are tissue regions with relevant information for survival analysis.

## 4. Conclusion and Future Work

In this work, we explore the notion of self-supervised learning and UDA for the identification of histological tissue types. Given the difficulty of obtaining expert annotations, we explore diverse UDA models in various label-scarce histopathology datasets. More importantly, we present a new label transferring approach from a partially labeled source domain to an unlabeled target domain. This is more practical than most previous UDA approaches tailored to fully annotated source domain data and/or tied to additional network branches dedicated to auxiliary tasks. Instead, we perform progressive entropy minimization based on the similarity distribution among the unlabeled target and source domain samples yielding discriminative and domain-agnostic features for domain adaptation. Through adaptation experiments, we show that our Self-Rule to Adapt method can discover the relevant semantic information even in the presence of few labeled source samples and yields a better generalization on different target domain datasets.

A future extension of this work is defining a self-supervised learning approach that can embed mixtures of tissues as publicly available datasets are solely composed of homogeneous patches. Such patches are not characteristic of the heterogeneity of complex images present in diagnosis routine and can lead to erroneous detections (e.g., background and stroma interaction interpreted as adipose). Furthermore, the segmentation achieved by our method can be used for clinically relevant applications, such as tumor-stroma ratio calculation, disease-free survival prediction or the adjuvant treatment decision-making.

## Acknowledgments

This work was supported by the Personalized Health and Related Technologies grant number 2018-327, and the Rising Tide foundation with the grant number CCR-18-130. The authors would like to thank Dr. Felix Mller for the annotation of WSI crops that greatly helped the evaluation of our method.

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

## Appendix A. Experimental Setup

In this section, we present the experimental setup. The architecture is first trained in an unsupervised fashion, and then as a second step, a linear classifier is trained on top as in Chen et al. (2020a). The architecture of the feature extractors, $f_\Phi$ and $f_\Psi$, is composed of a ResNet18 (He et al., 2016) followed by two fully connected layers (projection head) using rectified linear activation units (ReLUs) and with an output dimension of $D_{FC1} = 512$ and $D_{FC2} = 128$, respectively. We update the weights of $f_\Phi$ as $\theta_\Phi$ and $f_\Psi$ as $\theta_\Psi$ using standard backpropagation and momentum as described in Equation (6), respectively. We use $m = 0.999$ as momentum to update weights as described in He et al. (2020).

$$\theta_\Psi \leftarrow m\theta_\Psi + (1 - m)\theta_\Phi \tag{6}$$

The model is trained from scratch for $N_{epochs} = 200$ epochs using the stochastic gradient descent (SGD) optimizer (momentum $= 0.9$, weight decay $= 10^{-4}$), a learning rate $\lambda = 10^{-2}$ and a batch size of $B = 128$. For the similarity learning and easy-to-hard training, we set $\tau = 0.2$, $s_w = 0.25$ and $s_h = 0.2$. We apply random cropping, gray transform, horizontal/vertical flipping, and color jittering as data augmentations $\mathcal{T}_x$. Algorithm 1 presents the pseudo-code of our SRA method. For a fair comparison, we also use a ResNet18 backbone for the presented baselines. The classification performances are evaluated using a linear layer placed on top of the frozen feature extractor for $N_{epochs} = 100$ epochs using the SGD optimizer (momentum $= 0.9$, weight decay $= 0$), a batch size of $B = 128$, and a learning rate of $\lambda = 10$.

At each epoch, we sample $50,000$ example with replacement from both the source and target dataset to create a set $\mathcal{D}$ of $N = 100,000$ samples. We use 70% of K16 to train the unsupervised domain adaptation. The remaining 30% examples are used to test the performance of the linear classifier trained on top of the self-supervised model. We repeat this operation 10 times to obtain statistically relevant results.

## Appendix B. Detailed Overview of the Datasets

In this study, we use two publicly available datasets as well as an in-house cohort that contain patches of different tissue types found in the human gastrointestinal tract and that are extracted from H&E-stained WSIs. Figure 4 shows the occurrence and relationship of different tissue types across the datasets. Note that labels are not available for the in-house dataset.

---

**Algorithm 1:** Self-Rule to Adapt pseudo code

---

Initialize queue $\mathcal{Q}$ with normal distribution;
Normalize queue entries $q_i \in Q$;
**for** epoch $= 0$ **to** $N_{\text{epochs}} - 1$ **do**
    Create $\mathcal{D}$ by uniformly sampling from $\mathcal{D}_s$ and $\mathcal{D}_t$;
    Update easy-to-hard coefficient $r$ using Equation (5);
    **for** batch $\{x_i\}_{i=1}^B$ in $\mathcal{D}$ **do**
        Get augmented samples $\hat{x}, \hat{x}'$ from $\mathcal{T}_x$ ;
        Perform forward pass $z = f_\phi(\hat{x}),\ z' = f_\psi(\hat{x}')$ ;
        Normalize vectors $z, z'$ ;
        Compute in-domain loss $\mathcal{L}_{\text{IND}}$ using Equation (2);
        Calculate cross-domain matching $H^{\text{CRD}}$ using Equation (3) ;
        Determine easy-to-hard samples set $\mathcal{R}_{s/t}$ using Equation (5) ;
        Compute cross-domain loss $\mathcal{L}_{\text{CRD}}$ by replacing $\mathcal{D}_{s/t}$ with $\mathcal{R}_{s/t}$ in Equation (4);
        Compute $\mathcal{L}_{\text{SRA}} = \mathcal{L}_{\text{IND}} + \mathcal{L}_{\text{CRD}}$;
        Update $\Phi$ weights with backpropagation ;
        Update $\Psi$ weights with momentum using Equation (6);
        Update queue $\mathcal{Q}$ by appending $z'$;
    **end**
**end**

---

**Kather-16 (K16) Dataset**: The dataset (Kather et al., 2016) contains $5,000$ patches ($150 \times 150$ pixels, $74\mu m \times 74\mu m$) from multiple H&E WSIs. There are eight classes of tissue phenotypes, namely tumor epithelium, simple stroma (homogeneous composition, includes tumor stroma, extra-tumoral stroma, and smooth muscle), complex stroma (stroma containing single tumor cells and/or few immune cells), immune cells (including immune cell conglomerates and sub-mucosal lymphoid follicles), debris (including necrosis, erythrocytes, and mucus), normal mucosal glands, adipose tissue, and background (no tissue). The dataset is balanced with 625 patches per class.

**Kather-19 (K19) Dataset**: The dataset (Kather et al., 2019) consists of patches depicting nine different tissue types: cancer-associated stroma, epithelium, normal colon mucosa, adipose tissue, lymphocytes, mucus, smooth muscle, debris, and background. Each class is roughly equally represented in the dataset. In total, there are $100,000$ patches ($224 \times 224$ pixels) in the training set.

**In-house Dataset**: Our cohort is composed of 665 H&E-stained WSIs from our local CRC patient cohort. The slides originated from 378 unique patients diagnosed with adenocarcinoma and were scanned at a resolution of 0.248 MMP (40x). The WSIs are sampled to reduce the computational complexity of the proposed approach. From each WSI, we uniformly sample 300 ($448 \times 448$ pixels, $111\mu m \times 111\mu m$) regions from the foreground masks, creating a dataset with a total of $199,500$ unique, unlabelled patches. We assume that these randomly selected samples of our cohort are a good estimation of its tissue complexity and heterogeneity.

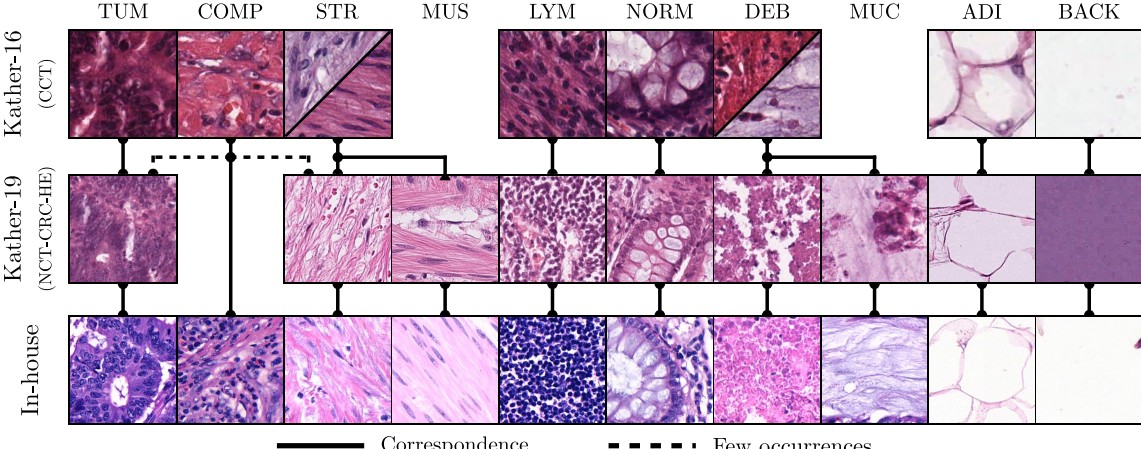

Figure 4: Examples images of the different tissue types present in the used datasets and their association. We use the following abbreviations: TUM: tumor epithelium, STR: simple stroma, COMP: complex stroma, LYM: lymphocytes, NORM: normal mucosal glands, DEB: debris/necrosis, MUS: muscle, MUC: mucus, ADI: adipose tissue, BACK: background. Examples from the in-house dataset are manually picked for comparison but are not labeled.

**Inconsistencies between K16 and K19**: An expert pathologist reviewed all three datasets to identify any potential discrepancies between the class definitions. We have identified the following issues:

- *Complex stroma:* The class is not represented in K19. However, few occurrences of the complex stroma are present in both the tumor and stroma class. Other samples are hard to distinguish and classify from regular stroma without context information.

- *Stroma:* In K16, the stroma class is a composition of stroma and smooth muscle. When performing domain adaptation, we consider the classes stroma and smooth muscle in K19 as a single stroma class to match the definition of K16.

- *Debris:* Similar to stroma in K16, the debris class is a mixture of multiple types of tissues. We observe examples of mucin, debris/necrosis, and loose tissue. For domain adaptation, we merge mucin into debris in K19. Note that collagenous tissue and blood are not present in K19, which is an additional example of an open set domain adaption.

## Appendix C. Self-supervision and the Importance of the Queue

In this section, we compare the performances of different self-supervised methods to the standard supervised learning approach when facing different levels of available data. The results are presented in Table 2. We report the performance of single domain classification on K16 and K19. The supervised approach uses ImageNet pre-trained weights. Self-supervised

Table 2: Results of classification of different self-supervised approaches with the supervised baseline on Kather-19 and Kather-16. We present the results for different percentages of available data. The top results for the are highlighted in bold. We use weighted F1 score.

| | Kather-16 Labels fraction | | | Kather-19 Labels fraction | | |
|---|---|---|---|---|---|---|
| Methods | 10% | 20% | 50% | 1% | 2% | 5% |
| Supervised[‡] | $85.8^{**}$ | $86.5^{**}$ | $87.9^{**}$ | $\mathbf{89.2}^{+}$ | $\mathbf{89.9}^{+}$ | $\mathbf{90.5}^{+}$ |
| SimCLR (Chen et al., 2020a) | $79.6^{**}$ | $78.9^{**}$ | $78.6^{**}$ | $76.9^{**}$ | $79.4^{**}$ | $80.7^{**}$ |
| SupContrast (Khosla et al., 2020) | $60.8^{**}$ | $73.2^{**}$ | $80.8^{**}$ | $78.7^{**}$ | $81.6^{**}$ | $85.0^{**}$ |
| MoCoV2 (Chen et al., 2020b) | $\mathbf{88.5}$ | $\mathbf{90.2}$ | $\mathbf{91.1}$ | $\mathbf{89.9}$ | $\mathbf{90.3}$ | $\mathbf{90.6}$ |

[‡] Model initialized with ImageNet pre-trained weights.
[+] $p \geq 0.05$; [*] $p < 0.05$; [**] $p < 0.001$; unpaired t-test with respect to the top result.

baselines are trained from scratch. For the classification results, we freeze the weights and add a linear classifier on top and train it until convergence. For SupContrast (Khosla et al., 2020) we jointly train the representation and the classification as described in the original paper.

We can observe that MoCoV2 (Chen et al., 2020b) outperforms the two other SOTA approaches. On K16 the model to gain up to 10% in terms of F1-score with respect to the other self-supervised baselines. In addition, MoCoV2 gives competitive results with the supervised baseline that is initialized with ImageNet weights. It proves that MoCoV2 is able to efficiently learns from unlabeled data to create relevant feature spaces. This mainly comes from the combination of the momentum encoder and the give access to a large number of negative samples.

## Appendix D. Ablation Study

We present the ablation study of our approach in Table 3. We denote $\mathcal{L}_{\mathrm{IND}}$ as the in-domain loss, $\mathcal{L}_{\mathrm{CRD}}$ as the cross-domain loss, and easy-to-hard (E2H) as the easy-to-hard learning scheme. For the baseline (no differentiation between in-domain and cross-domain), we consider the model where the training set $\mathcal{D}$ is the merged source and target domain data as in (He et al., 2020). Adding just the $\mathcal{L}_{\mathrm{CRD}}$ to the loss creates an unstable model, because we do not impose domain representation and thus the model converges toward incorrect solutions where random sets of samples are matched between the source and target datasets. $\mathcal{L}_{\mathrm{IND}}$ achieves a relatively good performances but fails to generalize knowledge to classes where texture differs (for example background). The introduction of the E2H procedure greatly improves the classification performances on debris and tumor classification while maintaining good performances on other classes.

The Figure 5 highlights the usefulness of the E2H scheme. Some tissue types might not have relevant candidates in the other set (open-set scenario). The example shown the figure is complex stroma (COMP), which is only present in K16 and not in K19. Without the E2H learning, the model tries to find matching candidates at any cost even if no suitable

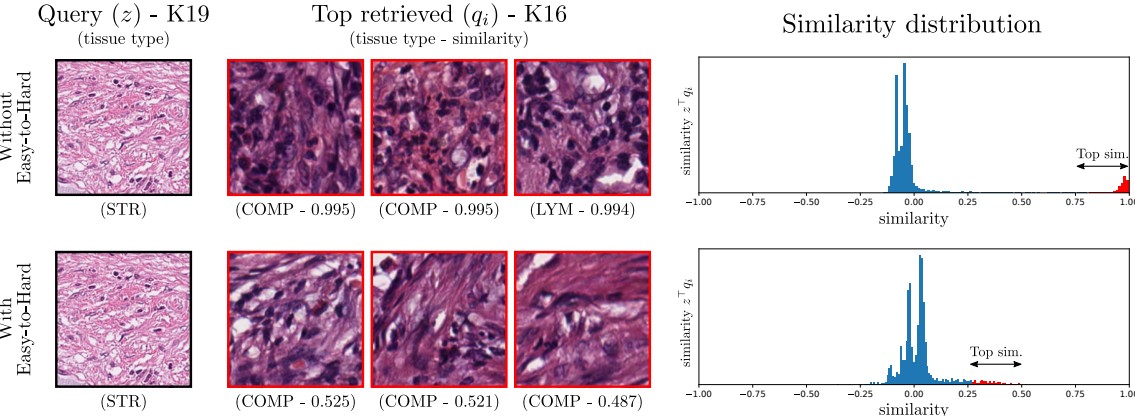

Figure 5: Effect on similarity distribution with (bottom) and without (top) E2H. Without E2H the model tries to optimize similarity for all queries at any cost and creates out-of-distribution samples (red). With E2H, the unpaired examples are still attached to the distribution (red).

Table 3: Ablation study for the proposed Self-Rule to Adapt (SRA) approach. We denote $\mathcal{L}_{\text{IND}}$ as the in-domain loss, $\mathcal{L}_{\text{CRD}}$ as the cross-domain loss, and E2H as easy-to-hard. We train the domain adaptation from Kather-19 to Kather-16. Only 1% of the source domain Kather-19 labels are used, and no labels for the target domain Kather-16. We report F1 and weighted F1 score for classes and average (all) respectively.

| Model | $\mathcal{L}_{\text{IND}}$ | $\mathcal{L}_{\text{CRD}}$ | E2H | TUM | STR | LYM | DEB | NORM | ADI | BACK | ALL |
|---|---|---|---|---|---|---|---|---|---|---|---|
| SRA$^\dagger$ | - | - | - | 36.8$^{**}$ | 45.4$^{**}$ | 27.1$^{**}$ | 30.8$^{**}$ | 45.2$^{**}$ | 43.1$^{**}$ | 43.6$^{**}$ | 38.9$^{**}$ |
| SRA | - | ✓ | - | 14.1$^{**}$ | 9.1$^{**}$ | 0.2$^{**}$ | 10.1$^{**}$ | 4.9$^{**}$ | 0.0$^{**}$ | 61.5$^{**}$ | 14.4$^{**}$ |
| SRA | ✓ | - | - | 88.1$^{**}$ | **72.8**$^{+}$ | 78.0$^{*}$ | 71.8$^{*}$ | 89.9$^{**}$ | 93.4$^{*}$ | 86.0$^{*}$ | 82.9$^{**}$ |
| SRA | ✓ | ✓ | - | 63.0$^{**}$ | 69.9$^{**}$ | **85.1** | 57.7$^{**}$ | **98.2** | **97.9** | 90.0$^{**}$ | 80.3$^{**}$ |
| SRA | ✓ | ✓ | ✓ | **93.4** | 72.9 | 82.7$^{*}$ | **67.9** | 96.5$^{*}$ | 97.0$^{**}$ | **97.2** | **86.9** |

$^\dagger$ Model jointly trained. Both source and target dataset are merged assuming a similar distribution.
$^{+}$ $p \geq 0.05$; $^{*}$ $p < 0.05$; $^{**}$ $p < 0.001$; unpaired t-test with respect to top result.

ones exist. This results in the occurrence of a subset of samples that have a near-perfect similarity to the query sample (top-right distribution plot, marked in red). Keeping the hyperparameter $r$ (Equation (5)) at a low level prevents the model from learning degenerated solutions (bottom-right distribution plot, marked in red). The same behavior is observed in other such open-set tissue classes (e.g., the absence of blood vessels and collagen in debris).

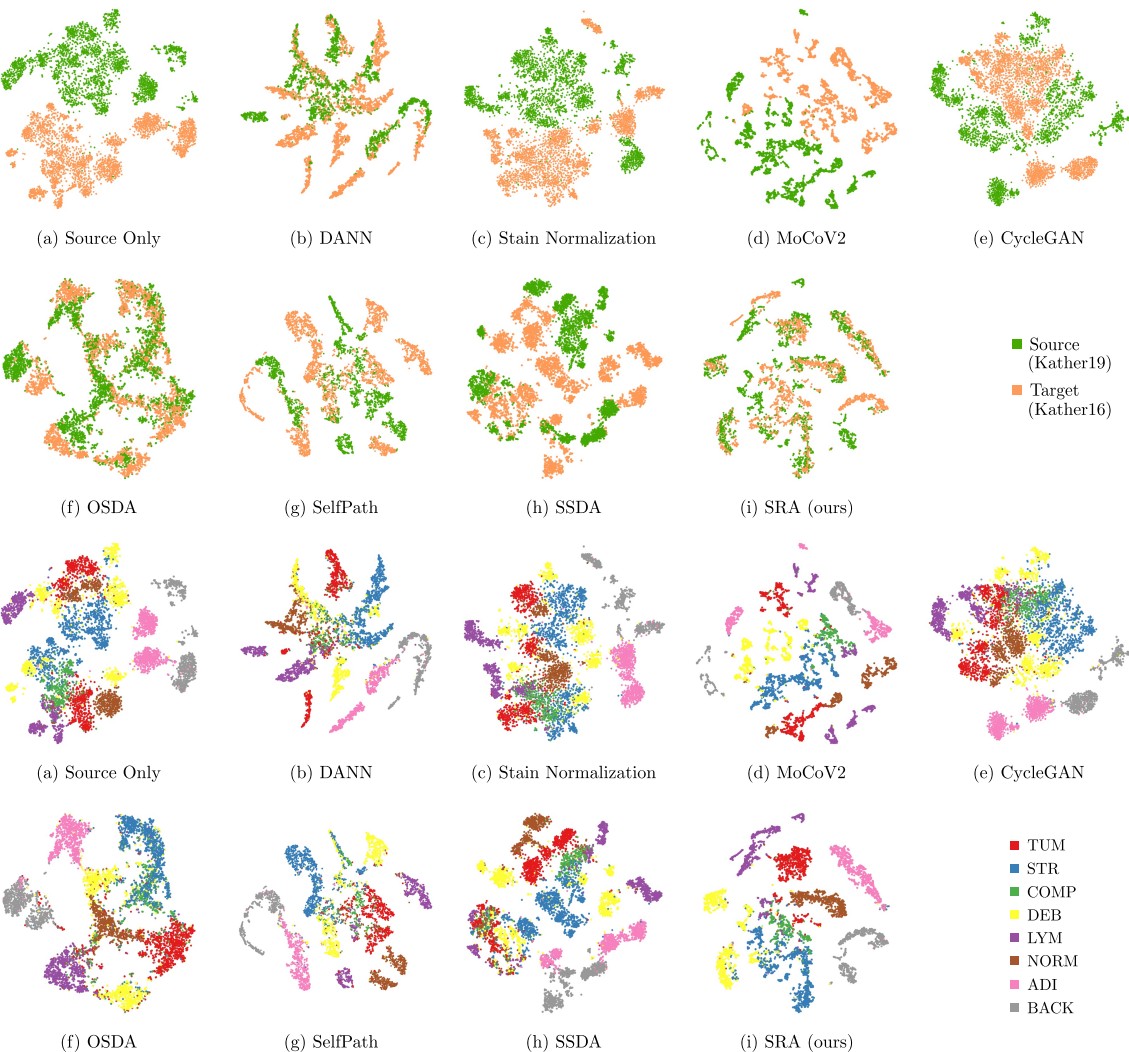

Figure 6: The t-SNE projections of the source (Kather-19) and target (Kather-16) domain embeddings. We show the alignment of the embedding spaces between the source and target domain for all presented models as well as the classes. The classes of Kather-19 are merged and relabeled according to the Kather-16 definition. The standard supervised approach is depicted in (a). We compared our approach (i) to other domain adaptation methods (b-h). Our approach (i) qualitatively shows the best alignment between the source and target domains.

## Appendix E.  t-SNE Projections

In this section, we display the complementary results to the ones presented in section 3.2. The embedding for all baselines and the presented approach are displayed in Figure 6. We show the alignment between the source (K19) and target (K16) embedding domain-wise as well as classes-wise.

