# OpenReview forum: "Self-Rule to Adapt: Learning Generalized Features from Sparsely-Labeled Data Using Unsupervised Domain Adaptation for Colorectal Cancer Tissue Phenotyping"
_MIDL.io/2021/Conference — MIDL 2021_

### Official Review · AnonReviewer4 · 2021-02-28

**Confidence:** 3
**Preliminary Rating:** 4
**Recommendation:** Oral

**Summary:**

The authors proposed a new self-supervised learning based framework for identifying histological tissue types. They introduced a new label transferring approach from a partially labeled source domain to an unlabeled target domain.  Through experimental results they showed the outperformance of their method compared to other domain adaptation methods.

**Strengths:**

The main advantage is the combination of both approaches: self-supervised learning and unsupervised domain adaptation. The novelty lies in proposing an entropy-based approach that progressively learns domain invariant features so that it boosts the robustness of the model to class definition inconsistencies and the presence of unseen tissue classes.

**Weaknesses:**

Further discussion of limitations of the model is highly needed and further explanation of training and testing steps are needed since it will help understand how to apply this framework in real-world problems.

**Deanonymize Review:**

no

**Justification Of The Preliminary Rating:**

The authors addressed an interesting problem with a novel self-supervised framework. They wrote the paper properly with strong argument. The supplementary materials are very interesting since they further show the reliability of the model.

**Paper Type:**

methodological development

**Special Issue:**

yes

---

> ### Author Response · Authors · 2021-03-15
> **Answer to AnonReviewer4**
>
> Thank you for your feedback. We added more explanations of the training and testing steps in the main text as it seems it was unclear without the help of the appendix. Due to space limitations, we cannot elaborate more on the robustness of the model in the main text. We included some additional results to motivate our selection of MoCoV2 as the baseline of the proposed approach and to highlight the limitation of other self-supervised approaches.
>
> The robustness of the model with respect to staining variation between whole slides acquired with different scanners is also an interesting discussion. We observed that our model as well as most basslines can suffer from large shifts in staining variation. Heavy data augmentation partially solves this issue. We plan to investigate this phenomenon by using a fixed set of slides scanned with different scanners to create a dataset of paired examples.

---

### Official Review · AnonReviewer3 · 2021-03-08

**Confidence:** 4
**Preliminary Rating:** 2
**Final Rating:** 3

**Summary:**

This paper presents a self-supervised learning-based method for the identification of histological tissue types under the unsupervised domain adaptation setting. Different from the traditional UDA, the proposed method utilizes a small number of the labeled source domain images. The key idea is to adopt the momentum conservative learning framework and design an in-domain loss and cross-domain loss to guide the network learning. The proposed method outperforms baselines for domain adaptation of colorectal tissue types and further validates the proposed approach on the in-house clinical cohort.

**Strengths:**

1. The sparsely-labeled UDA setting is important in medical image analysis domains.
2. The proposed method achieves good performance on the public datasets and the authors also evaluate the proposed method on in-house clinical datasets.


**Weaknesses:**

1. The technical novelty is limited. As mentioned by the author, the main idea of the proposed method is very similar with the previous work.

Kim, D., Saito, K., Oh, T.H., Plummer, B.A., Sclaroff, S. and Saenko, K., 2020. Cross-domain self-supervised learning for domain adaptation with few source labels. arXiv preprint arXiv:2003.08264.

2. The presentation and motivation of "cross-domain loss" is unclear. Actually, it is a little difficult for me to understand this part.


**Deanonymize Review:**

no

**Detailed Comments:**

1. How do you identify the source queue samples and target queue samples when calculating the loss? Do you use an extra indicator?

2. Please further clarify the difference between the proposed method and the related method (Kim et al.)


**Final Rating Justification:**

I have read the rebuttal. The authors further clarify the difference between their method and Kim et al. and I also agree that it is important to keep a good balance between novelty and clinical application. Therefore, I change my rating to weak accept. I suggest the authors further clarify the difference with Kim et al. in the final version.

**Justification Of The Preliminary Rating:**

In general, this paper is easy to follow and the authors conduct extensive experiments to show the effectiveness of the proposed method. My main concern with this submission is that the key technical part of this method is very similar to the previous one, as the in-domain loss and cross-domain loss are identical to the ones in Kim et al. If the authors can clarify the difference with the previous related works and compare with them, I will consider changing my preliminary rating.

**Paper Type:**

methodological development

**Questions To Address In The Rebuttal:**

See weakness and Detailed comments.

**Special Issue:**

no

---

> ### Author Response · Authors · 2021-03-15
> **Answer to AnonReviewer3**
>
>
> Thank you for your feedback and comments. We will try our best to answer your concerns:
>
> - Allow us to discuss the similarity of our method and the work of Kim et al. 2020 (Cross-domain self-supervised learning) so that we can highlight them.
>     - Firstly, we introduce an easy-to-hard learning scheme that is not present in Kim et al. 2020. This additional feature is mandatory as it avoids the model's collapse toward trivial solutions. For example, one can imagine that the model would converge to a poor representation/matching of the source queries in the target space, but as long as other samples representation are "poorer" the loss would be satisfied (App. D, Table 3). Such degenerated solutions are not possible with our easy-to-hard procedure as we first rely on samples with high confidence (i.e., low entropy). Moreover, we never enforce strict matching for samples with high entropy. As a result, samples from the target domain that do have matching queries are not linked to irrelevant source samples.
>     - Secondly, the work of Kim et al. 2020 work is tied to the definition of the two memory banks that might be heavy to maintain. For each element of the batch, we need to compute the similarity of the sample with all elements of the memory banks. When dealing with whole slide images we can benefit from multiple hundreds of unique samples which, in this case, is a limitation. In our case, we use a queue instead of a memory bank, which allows us to train on as many unique samples as we wish without needing to keep track of the embedding of all previous samples.
>     - Thirdly, Kim et al. 2020 define the similarity of in-domain samples with respect to the corresponding memory bank (e.i., the similarity of the source sample with the source memory bank). Here, we choose to compute the in-domain similarity between augmented samples present in the batch. This approach has the advantage of allowing us to make our model more robust to staining and color variation as we optimize their paired similarity. The augmented samples embeddings $z_i$ and $z_i'$ are directly compared together (equation 1) which greatly helps contrastive learning and makes it a challenging (yet efficient) task with the help of random cropping. We will try to make this difference more clear in the text.
> - Without the cross-domain loss, the optimization of the model only relies on the in-domain loss. As a result, the model will optimize the loss and create distinct distributions of both the source and target domains (see Appendix C, Fig.6 (d) MoCoV2). Those distributions are not aligned and then when the model (trained on the source labels) is applied to the target domain it will create discrepancies. Here, the cross-domain loss encourages the alignment of the feature distributions (see Appendix C, Fig.6 (i) SRA) to avoid this phenomenon.
> - Indeed an index is maintained to identify and differentiate the queue samples belonging to the source from those belonging to the target domain. In figure (Fig. 1) the indexation is represented with the different colors (orange for source and green for target).
> - In addition, we would like to emphasize that we tried to keep a good balance between novelty and clinical application. Here, we also show that recent work can be adapted, improved, and applied to a real-world scenario with whole slide images and thus are not limited to theoretical applications. Segmentation is an important up-stream task for many analyses in digital pathology, e.g. identification of mucinous tumors, tumor-infiltrating lymphocytes (TILs) identification.
>
> We hope we have addressed all of your questions and concerns to your satisfaction.

---

### Official Review · AnonReviewer2 · 2021-03-08

**Confidence:** 4
**Preliminary Rating:** 3
**Recommendation:** Poster
**Final Rating:** 3

**Summary:**

This paper proposes an unsupervised domain adaptation method for colorectal cancer tissue phenotyping. The proposed method uses a contrastive learning based in-domain loss to learn the source and target domain features respectively, and an entropy minimization based  cross-domain loss to encourage domain alignment with easy-to-hard learning. Experiments are conducted on both public and in-house datasets with different tasks.

**Strengths:**

- The proposed method seems reasonable, though the cross-domain loss should be explained more clearly.
- Experimental evaluation is conducted on two public datasets for patch classification and in-house data for WSI segmentation.

**Weaknesses:**

- The motivation of the introduced cross-domain loss is not clear. It is difficult to understand the rational that “if we embed a random sample drawn from D_s we expect to be able to find a limited number of candidates in D_t”. The reason about why the entropy minimization can reduce the domain mismatch is not clearly explained.

- The concept of “few labeled source samples” is a bit vague. It would be better if the authors could include the comparison results with other methods when the number of labeled source samples increases, to show the benefits of the proposed method when the labeled source samples are limited.

- For the validation on WSIs, only results of three examples are presented. Why not present the summarized results for the whole dataset?

- How the labeled source domain data are used in the framework is not described, such as how the supervised loss are combined with the unsupervised losses in the training process.


**Deanonymize Review:**

no

**Detailed Comments:**


- The ablation study and in-depth analysis of the presented method is important and should be better included in the main body instead of the appendix.

**Final Rating Justification:**

I have read the authors' response and part of my concerns have been addressed, I thus maintain my rating as weak accept.
The authors explain the cross-domain loss more clear in the response, but the results with different label percentages are for training single domain classifiers rather than the cross-domain learning. The authors should also update the main text to make the cross-domain loss and training details more clear.

**Justification Of The Preliminary Rating:**

The authors try to address an important problem with a properly-designed method. Evaluation on different datasets and tasks validate the method. However, there are still major concerns on the method and experiments which need to be addressed by the authors. Please see the comments in “Weaknesses”.

**Paper Type:**

methodological development

**Special Issue:**

no

---

> ### Author Response · Authors · 2021-03-15
> **Answer to AnonReviewer2**
>
> Thank you for your valuable feedback. We will try to address your concerns as well as possible.
>
> - Regarding “if we embed a random sample drawn from $D_s$ we expect to be able to find a limited number of candidates in $D_t$”. Let us try to illustrate this with a toy example. Assuming we have two datasets where we know we have a limited number of classes (e.g. "apple", "pear" and "orange"). When matching a sample of "apple" from the source domain we should only retrieve the similarly looking "apple" candidates in the target domain while leaving out "pear" and "orange" candidates. If the computed similarity is high only for the matching candidates (here "apple") and not the other candidates the entropy would be small (equation 3). However, if the similarity is high for all candidates ("apple" is equally similar to "apple", "pear", and "orange") the entropy would be large as the source sample is not well defined in the target domain. This principle can be extended to our task.
>
> - Some preliminary results were computed with different label percentages when training single domain classifiers. The results were not included in this work in the first place to keep the focus on cross-domain learning. We compared the supervised approach to X. Chen et al. 2020 (MoCoV2), T. Chen et al. 2020 (SimCLR), and Khosla et al. 2020 (Supervised Contrastive Learning) with different label quantities. We updated the supplementary material to include those results.
>
> - Unfortunately, these were the only three slides with available annotations. We had to ask an expert pathologist to label them, which was very time-consuming. All other slides are not labeled which is also part of the motivation of our work. We wanted to avoid labeling all the data. However, we selected relevant regions to present qualitative results.
>
> - In this work, we first train the model in a self-supervised fashion with our SRA loss. Then, in a second step, we freeze the weights of the model and train a linear classifier on top of it with only a subset of labels (e.i., 1%). The mentioned details were only present in the appendix. We updated the main text to make this more clear.
>
> We hope we have fully addressed all of your questions and concerns.

---

### Official Review · AnonReviewer1 · 2021-03-09

**Confidence:** 3
**Preliminary Rating:** 3
**Recommendation:** Oral

**Summary:**

This paper adapts the recent Momentum Contrast (MoCo) method for self-supervision to the problem of unsupervised domain adaptation. The authors embed domain adaptation within the self-supervised learning scheme, enabling them to learn a classifier on a target domain using few labelled source examples. The authors validated and compared their method against various competing baselines showing strongest performance and approaching the upper bound (training a network on the target dataset with labels).

**Strengths:**

1. This is a nice method - extending the self-supervision framework of Momentum Contrast to operate in the challenging scenario of unsupervised domain adaptation framework is clever and very interesting

2. The results are impressive and the qualitative evaluation (t-SNE) nicely illustrates that the objectives stipulated in the loss function indeed learn similar features across domains

**Weaknesses:**

1. The presented losses are very similar to Kim et al. 2020. Is the main novelty of the paper the idea to introduce MoCo with the losses from Kim et al.? As far as I can tell, Kim et al. use 2 feature encoders whilst in the proposed work, the authors use an encoder and a momentum encoder to deal with larger batches. Is this the key difference and main novelty? A better explanation is required. It is not sufficient to simply state "Motivated by Ge et al. and Kim et al., we extend the domain...". As a result, it is difficult to ascertain whether the proposed losses are fully novel or not.

2. I cannot find any other major weaknesses as most queries I had when reading the manuscript were subsequently answered in the appendix. I would perhaps like to see a better description as to why Momentum Contrast (MoCo) was chosen and why MoCo is successful in learning good visual representations. Was it only chosen because of the momentum encoder that allows a large dictionnary of samples? Further to this, I would also like the potentially see the pseudo-algorith in the main text as I feel this would aid reading comprehension. However, this is stylistic and a minor comment.


**Deanonymize Review:**

no

**Detailed Comments:**

1. In the introduction. "It involves a two-step training scheme..and to create supervision from itself". Only one-step is described and the statement is thus incomplete.

2. I would cite van den Oord [https://arxiv.org/pdf/1807.03748.pdf] when introducing Equation 1.

3. The method effectively looks to match the empirical marginal distributions of both the source and target. Is the method extendable to cases where you have more than one source?

4. Did you try other ways to formulate cross-domain loss in your experiments? In the current formulation, you have one query sample and calculate the entropy with respect to store queue samples. Did you try an MMD loss for instance by passing a batch of samples through the encoder to then minimise MMD(z, q)?

**Justification Of The Preliminary Rating:**

I've decided to give a preliminary rating of Weak Accept. This is a strong submission. It is an interesting extension of the some of the state of the art in self-supervision with domain adaptation with strong validation/results.

**Paper Type:**

methodological development

**Questions To Address In The Rebuttal:**

I will be happy to raise my score to Strong Accept if the authors can address my issues on similarities to Kim et al. helping to outline the main contribution of the work and its novelty.

**Special Issue:**

yes

---

> ### Author Response · Authors · 2021-03-15
> **Answer to AnonReviewer1**
>
>
>
> Thank you for your detailed feedback and comments on our work. We will address all inputs individually:
>
> - The presented loss is indeed inspired by Kim et al. 2020 work. The difference lies in the modification of their approach to be less computationally demanding and more robust to outliers. As mentioned, they rely on 2 encoders and 2 memory banks (source and target). When working with whole slide images, the target domain can contain hundreds of samples. It's even more demanding as for each sample you have to compute similarity with all samples from one of the memory banks. In our case, the size of the queue can be adapted while keeping the number of whole slide image samples large. The second difference is that for the in-domain loss we optimize the similarity between the augmented samples ($x_i$ and $x_i'$) and not with respect to the memory bank/queue as in Kim et al. 2020. This allows our model to strongly benefit from data augmentation as described in T. Chen et al. 2020 (SimCLR). Therefore, data augmentation can be designed to ensure stain and rotation invariance for example. Moreover, the introduction of the easy-to-hard learning scheme is not present in Kim et al. 2020 and greatly helps the model to converge and avoid trivial solutions. For example, without easy-to-hard learning, the model could force all target samples to have a matching pair in the source domain. However, this is usually not the case. This situation is depicted in Appendix D, Fig. 6. We tried to revise some sentences in the document to better highlight the differences between those two works.
>
> - Before selecting MoCoV2 we tried different approaches such as T. Chen et al. 2020 (SimCLR), and  Khosla et al. 2020 (Supervised Contrastive Learning). MoCoV2 is the best performing among them when training with a reduced batch size (B=128) and few available labels. Our findings are that the use of the queue is of great help when training our algorithm as the availability of negative samples is key for contrastive learning. Unfortunately, these results cannot be included in the main text due to space limitations. However, we updated the supplementary material (Appendix C) to include them.
>
> - Indeed, the two-step training scheme described in the main text was not very clear without the help of the appendix, we updated the main text to make it more precise.
>
> - The method can be extended to multiple sources and even multiple targets, which is an idea for future work. Equation (4) would need to be updated to include multiple source/target domains. This could be helpful if we have multiple source datasets from online cohorts.
>
> - MMD as described in Hongliang Yan et al. 2017 (for example), was considered but not implemented as computing the presented baseline was already time-consuming. However, it could be a great additional baseline for comparison in future work.
>
> We hope we have fully addressed all of your questions and concerns.

---

### Meta-Review · Area_Chair1 · 2021-03-29

**Recommendation:** Accept (Oral)

**Metareview:**

The authors were able to resolve some of the reviewers' initial concerns. The paper can be of interest to the community and presents promising results.

**Paper Type:**

methodological development

---

### Decision · Program_Chairs · 2021-03-31

**Decision:**

Accept

**Comment:**

Congratulations your paper has been selected as long oral.